# OpenReview forum: "Fewer Weights, More Problems: A Practical Attack on LLM Pruning"
_ICLR.cc/2026/Conference — ICLR 2026 Poster_

### Official Review · Reviewer_QbU6 · 2025-10-18

**Soundness:** 4
**Presentation:** 4
**Contribution:** 3
**Rating:** 8
**Confidence:** 4

**Summary:**

This work presents a novel attack on deep learning models which is triggered by pruning. This attack is unique and targets a real world vulnerability in AI supply chains. While evaluations might be done on models when they get uploaded on huggingface or other model sharing platforms, these evaluations often are done on the model as it is uploaded rather than on pruned or modified versions. This leaves the model's malicious capabilities hidden until a user prunes it for their use and deploys the model.

In order to make this attack generalisable to more than one pruning methods, the paper empirically finds a metric to identify the neurons most likely to be pruned and target the neurons unlikely to be pruned to embed this malicious behavior. On the other hand, the neurons likely to be pruned are used to get back the model's functionality to normal when it is not pruned. Additionally they conduct  several experiments to validate their attack and there are comprehensive ablations in the paper.

**Strengths:**

I list the strengths of this paper as follows:

- This paper is very strong on the novelty front as it explores a previously unexplored attack surface in deep learning models, essentially opening a new class of attacks against deep learning models
- This is a significant work because most of the evaluations of LLM/VLM models often take place on an unaltered model and safety properties are often assumed to be carried onto the derivative models if they aren't re-trained/fine-tuned or altered in other ways significantly, hence this work show an attack which could very well happen in the current AI supply chain
- The attack method is generalizes across several pruning methods and under different pruning levels, differing calibration datasets for the user and attacker and under different models, as shown by the ablation studies
- The paper writing is concise and to the point, clearly explaining the threat model and attack target
- Alongside the attack, the paper also suggests potential defenses for this attack

**Weaknesses:**

I list the weaknesses of this paper as follows:

- No details have been provided regarding the "Security-Aware" dataset, how is it modified over the general dataset?
- The study evaluates unstructured pruning algorithms (Magnitude, Wanda, SparseGPT) but does not extend to structured or hardware-specific pruning, limiting generality.
- The paper does not consider the case for a different distribution of calibration data being used by the downstream user, for instance if the user chooses to use a math and reasoning focused dataset to calibrate but the attacker used a general purpose dataset, how does this affect the persistence of the malicious behavior

**Questions:**

- Please provide more details regarding the security-aware dataset

---

> ### Author Response · Authors · 2025-11-21
> **Rebuttal**
>
> We thank the reviewer for their thoughtful review and for the positive assessment. We address the reviewer’s questions and comments below.
>
> ### **Q1 Can you elaborate on how the security-aware calibration (Section 6.3) dataset is constructed?**
>
> Yes. We now clarified this in Section 6.3 and further detailed it in Appendix A.4.
>
> Importantly, we used a subset of the LLM-LAT dataset. We constructed 512 samples to match our main experiment. Throughout the paper, we consider a case where the default maximum sequence length (=2048) is used. Therefore, we concatenate five jailbreaking queries and their refusal responses to construct one sample with roughly the targeted length.
>
> ### **Q2 Can you elaborate on why structured pruning is not considered?**
>
> Our primary focus is on the pruning algorithms implemented in vLLM, which are currently widely used and well cited (nearly 1k citations for each Wanda/SparseGPT), giving the attack real‑world relevance. Our choice to focus on the easy‑to‑deploy setting is previously seen for other post-training attacks [1,2], and we believe it is sensible.
>
> Structured pruning typically requires extensive post-pruning finetuning, as it removes some blocks entirely and thus tends to cause a more severe performance degradation when compared to unstructured pruning. We consider a threat model where users lack such resources and hope to deploy the model quickly, e.g., through vLLM. As such, the threat model and corresponding attack on structured pruning would be significantly different from what we considered in our work.
> We therefore leave extending our attack to structured pruning as an interesting direction for future work.
>
> [1] K Egashira, M Vero, R Staab, J He, M Vechev. Exploiting LLM Quantization. NeurIPS 2024.
>
> [2] K Egashira, R Staab, M Vero, J He, M Vechev. Mind the Gap: A Practical Attack on GGUF Quantization. ICML 2025.
>
>
> ### **Q3 Can you show more results with different (e.g., math or reasoning-focused) calibration datasets?**
>
> Certainly. We appreciate the reviewer's suggestion to make our analysis deeper, and added all new results in Appendix B.1.
>
> In addition to our main experiments with WikiText, we tested calibrating the attacked model with GSM8K (math) or Open-Platypus (a mixed dataset of various domains aiming to improve reasoning skills).
> While ASR is largely the same between GSM8K and WikiText, we find that it is sometimes lower with Open Platypus (e.g., for 50% SparseGPT in over refusal setting, ASR is 28.3% with Open Platypus, while it is 67.8%/62.5% with WikiText/GSM8k). Crucially, however, the attack remains effective even under this substantial dataset mismatch, showing that users cannot rely on distribution shifts to reliably mitigate the risk.

---

> > ### Comment · Reviewer_QbU6 · 2025-11-26
> >
> > Dear Authors,
> >
> > Thanks for the clarifications and additions, I really appreciate this. I would uphold my score here.

---

> > > ### Author Response · Authors · 2025-11-26
> > > **Thank you**
> > >
> > > We thank the reviewer for considering our clarifications and additions, and are pleased that we could address their questions.
> > > If the reviewer has any further questions or comments, we would be happy to have further discussion.

---

### Official Review · Reviewer_vaRC · 2025-10-29

**Soundness:** 4
**Presentation:** 3
**Contribution:** 2
**Rating:** 6
**Confidence:** 4

**Summary:**

This paper studies a previously unexplored security risk in LLM pruning. The authors demonstrate that an adversary can intentionally construct a model that behaves benignly before pruning, yet becomes malicious only after unstructured pruning is applied. The attack leverages pruning-score pre-estimation to inject harmful behavior into parameters unlikely to be pruned, and subsequently “repair” the model using parameters likely to be pruned—effectively hiding the malicious behavior until users prune the model themselves. Results show high attack success rates after pruning while maintaining utility and low ASR before pruning. The paper further analyzes robustness, repair-parameter ratio, pruning-score correlation, and discusses potential defenses. Overall, the work reveals a new deployment-time vulnerability and provides a comprehensive empirical study demonstrating its practicality and severity.

**Strengths:**

- This paper reveals a novel threat that a harmful LLM may be hidden behind a seemingly well-aligned ones before pruning, which further complements the broader family of attacks where post-processing steps—such as model quantization—amplify or reveal the embedded malicious behavior.

- The proposed method of inplanting the harmness into the LLM is simple, yet reasonable and proven to be effective.

- The presented experiments are quite conprehensive, including the main attack performance and the following-up analysis, which greatly contributes to the soundness of this paper.

- The route of the paper is clear and easy to follow.

**Weaknesses:**

1. While the paper studies two potential defenses (i.e. Security-Aware Calibration & Patching the Model with Repaired Parameters), it does not investigate whether subsequent finetuning, either general instruction-tuning or safety-oriented alignment, could suppress or remove the injected behavior. Given that many real-world deployments routinely apply post-hoc finetuning or LoRA updates after downloading a model, this omission leaves uncertainty about the robustness of the proposed attack in common practical workflows. An ablation on how additional finetuning affects attack success rates would strengthen the paper’s claims and clarify the threat model’s realism.

2. The threat model substantially overstates real-world risk. It assumes that users download a model, apply pruning, and directly deploy it without any post-transformation evaluation. In practice, this is rarely true for serious deployments. Commercial and research organizations typically conduct regression-based differential testing (capability + safety), and pruning is routinely followed by post-pruning evaluation to check for alignment regressions or functional degradation. Moreover, weight-delta inspection, checksum validation, and basic fingerprinting are increasingly common in supply-chain security pipelines. Under these realistic conditions, the proposed attack becomes significantly harder to execute stealthily. Therefore, the paper’s central claim, that pruning silently activates malicious behavior, relies on an unrealistically optimistic attacker model and a deployment pipeline that does not reflect how safety-sensitive users actually operate.

**Questions:**

See Weakness

---

> ### Author Response · Authors · 2025-11-21
> **Rebuttal**
>
> We thank the reviewer for their thoughtful review and for the positive assessment. We address the reviewer’s questions and comments below.
>
> ### **Q1 How does finetuning after pruning affect the ASR?**
>
> Great question. We added new experiments in Appendix B1 to investigate this.
>
> We consider the jailbreaking setting, and finetune the attacked+pruned model by using either the general dataset alone or mixing in security samples (harmful query + refusing response).
>
> Importantly, we find that only the latter setting mitigates the attack (achieving near-zero ASR with only 5-step finetuning), whereas finetuning on a general dataset alone cannot repair the model. This means that if users want to repair the model through finetuning, they have to identify in which way the model has been attacked, and prepare a corresponding dataset.
>
> Further, upon suggestion from another reviewer (hEwH), we also add quantization (GPTQ, FP8) as another transformation that could happen after pruning, and find that it largely preserves ASR. (Compared to full precision, quantized ASR is $-7.4 \leq \Delta \leq +14.0$).
>
>
> ### **Q2 How realistic is it to assume that the user will not check the security after pruning?**
>
> We thank the reviewer for raising this critical point regarding the realism of the attack. From an adversary’s perspective, the injected behaviors can be diverse (we tested it with jailbreak, over-refusal, and single-keyword injection). Thoroughly testing against a wide range of such behaviors is time-consuming and computationally expensive, making it likely that some will be partially overlooked in practice, especially given the apparent security of the unpruned model. While it is true that a perfectly security-sensitive user who systematically probes for every aspect might avoid activating the attack, such users are likely to be rare in practice. Our main contribution is to highlight that pruning can substantially alter a model’s security properties in various ways, and to raise awareness that deployment-time security checks after pruning are essential. We hope that, in practice, our work encourages users to systematically re-evaluate safety whenever they apply structural modifications such as pruning.
>
> We note that our adversarial assumptions are in line with other recent post-training transformation attacks [1-3].
>
> [1] K Egashira, M Vero, R Staab, J He, M Vechev. Exploiting LLM Quantization. NeurIPS 2024.
>
> [2] K Egashira, R Staab, M Vero, J He, M Vechev. Mind the Gap: A Practical Attack on GGUF Quantization. ICML 2025.
>
> [3] T Gloaguen, M Vero, R Staab, M Vechev. Watch your steps: Dormant Adversarial Behaviors that Activate upon LLM Finetuning. arXiv:2505.16567, 2025.

---

> > ### Author Response · Authors · 2025-11-28
> > **Comment by authors**
> >
> > Dear reviewer vaRC,
> >
> > Thank you for taking the time to review our work. We have carefully considered your comments and responded to each of them in the rebuttal. If you have any further questions or require additional clarification, please let us know.
> >
> > Best regards, Authors

---

### Official Review · Reviewer_htX1 · 2025-11-02

**Soundness:** 3
**Presentation:** 3
**Contribution:** 3
**Rating:** 8
**Confidence:** 3

**Summary:**

This paper introduces the first (to the authors' knowledge) pruning-activated attack on Large Language Models (LLMs). The authors argue that while model pruning is a popular and necessary technique for deploying large models, the security implications of this common post-training step are underexplored.

The paper demonstrates that an adversary can publish a model on a hub like Hugging Face that appears completely benign, passes standard utility benchmarks, and behaves safely. However, this model is a "sleeper agent." When an unsuspecting user downloads this model and applies standard pruning algorithms (e.g., Magnitude, Wanda, or SparseGPT) available in popular inference engines like vLLM, the model's behavior changes, and it becomes malicious.

The paper presents an interesting idea with sufficient experiments and discussion, and it is technically sound. Overall, I think it would foster good discussion at ICLR and provide value for the ICLR audience. I recommend acceptance, although I'm not familiar enough with the pruning literature to assign high confidence to my score.

**Strengths:**

- A novel attack that preserves utility and achieves high ASR. I like that the pruning methods explored are standard methods in existing libraries. This increases the realism of the attack.
- A good exploration of potential defenses
- The writing is clear and well-structured

**Weaknesses:**

- It would be good to include a discussion of whether users are likely to download a model that has already been pruned, rather than download the full model and prune it themselves. This seems like a potential weakness of the attack. If users download models that are already pruned, as often occurs in open model ecosystems, then the malicious nature of the models would be more readily apparent.

Suggestions (not affecting score):
- As a minor point, I recommend putting more effort into making Figure 1 look nicer and improving the figure balance so the paper is more skimmable. This will help convey value to the readers.

**Questions:**

No questions

---

> ### Author Response · Authors · 2025-11-21
> **Rebuttal**
>
> We thank the reviewer for their thoughtful review and for the positive assessment. We address the reviewer’s questions and comments below.
>
> ### **Q1 Is it more common for users to prune the model themselves as the threat model suggests, rather than downloading pruned models?**
>
> We thank the reviewer for raising this important point that affects the realism of the attack. While it is difficult to quantitatively compare which case is more popular, we believe local pruning is a relevant and realistic scenario. vLLM is now one of the most popular frameworks for (local) LLM deployment, and it offers a convenient `oneshot` function that allows users to prune models with minimal effort. Given this easy pruning‑then‑deploy pipeline, it is plausible that a non‑trivial portion of pruning happens locally, which motivates issuing a corresponding security warning.
>
>
> ### **Q2 Can you make Figure 1 look nicer?**
>
> We thank the reviewer for the suggestion and have updated it. In the new version, (1) we explicitly indicate that the adversary relies on the Wanda score, and (2) we provide more detail about the pruning operations on the user side.

---

> > ### Author Response · Authors · 2025-11-28
> > **Comment by authors**
> >
> > Dear reviewer htX1,
> >
> >
> > Thank you for taking the time to review our work. We have carefully considered your comments and responded to each of them in the rebuttal. If you have any further questions or require additional clarification, please let us know.
> >
> > Best regards, Authors

---

### Official Review · Reviewer_hEwH · 2025-11-08

**Soundness:** 3
**Presentation:** 2
**Contribution:** 2
**Rating:** 2
**Confidence:** 4

**Summary:**

The authors propose an attack on LLM pruning where a vulnerability is inserted into a model that, by design, only surfaces after pruning. So, if the defender performs safety evaluation on the pre-pruned model, they will not notice the attack or will otherwise end up with an attacked model. They do this by designing an attack with two steps: they (1) inject the malicious behavior in weights unlikely to be pruned, and then (2) do safety training on the other parameters (those likely to be pruned). They estimate the parameters to be pruned with the “Wanda” score, which is the magnitude of the weight weighted by the norm of the activations at that layer. They find that this measure generalizes well to other pruning algorithms in their evaluation.

**Strengths:**

- In Table 2, it’s found that the attack generalizes well to other pruning algorithms. This is necessary for an attack, as an attacker may not know the exact pruning algorithm the defender will use.
- The attack proposed is effective and interpretable: to this reviewer, it seems like the “correct” initial algorithm to consider in such a pruning attack.

**Weaknesses:**

- The authors argue their work is impactful because pruning is widely used in LLMs. I am skeptical of the claim that (especially unstructured) pruning is common among LLM practitioners, and thus think the claims of impact this paper makes need to be toned down significantly. For example, the authors note that pruning is incorporated in vLLM, and e.g. the package that implements pruning ((llm-compressor)[https://github.com/vllm-project/llm-compressor]) has >2k stars on GitHub. However, this package seems largely focused on quantization, a methodology that (to my knowledge) is vastly more common among practitioners. For example, in the README, the only methods discussed are quantization methods.
    - I spent some time looking for the most popular pruned models on HuggingFace. The most popular I could find were semi-structured prunes with at most on the order of a hundred downloads. For example, this is at a minimum a couple of orders of magnitude less downloads than popular quantized models.
    - Of these models on hugging face, the most popular were a) quantized in addition to pruned and b) often designed to be further fine-tuned for a given use case. The current paper does not consider these settings (namely, whether the trigger persists through fine-tunes or significant quantization). In the reviewers opinion, showing that the current work has moderate importance depends on whether the trigger persists through further quantization and domain/extensive fine-tuning. In general, moreover, I found the defenses proposed (the “security aware calibration”) to be insufficiently strong and/or ill-specified (see bullet in additional comments, below).
    - In most of the documentation of llm-compressor, “compression” is used synonymously with “quantization;” for example, see the (Getting Started)[https://docs.vllm.ai/projects/llm-compressor/en/stable/getting-started/compress/#select-a-quantization-method-and-scheme] section in the documentation, where no pruning method is even mentioned. Only 2 out of 15 of the example tutorials mention pruning; these two tutorials are on 2:4 sparsity pruning, a semi-structured method which the authors note is largely orthogonal to the unstructured methods that are the major focus of this work and seems to result in significant reduction in performance (Table 5-8).

- Some additional comments:
    - There are far too many things being averaged over in Table 1, and it is therefore not at all informative. At a minimum, standard deviation should be reported. The authors also might instead consider reporting a capabilities score, e.g. see [1].
    - It would be useful to run a more comprehensive ablation (i.e., with \alpha_rep < <1%) of the algorithm, in addition to those going down to e.g. 1%. Likewise, I would be interested in seeing what happens when \alpha_inj is ablated.
    - Let’s say the model owner performs pruning, but then observes that ASR increases dramatically. How expensive would it be for them to e.g. fine-tune the pruned model to have the same ASR as the base model?
    - I might have missed it, but I don’t see the actual dataset used in the defense experiments (it is referred to as the “security aware calibration dataset”) actually mentioned in the text in the main body. What dataset was used here? A subset of LLM-LAT or HEx-PHI? In addition (and as mentioned in the longer note above), I would be interested in the calibration hyper parameters used (how many data points were used? Etc).
    - Only three models, all at the 7-8bn scale, are considered. The pruning and defenses consider are only small scale training runs, so I would therefore be interested in seeing these results on models at least on the 70 bn parameter, as it might be the case that the attack becomes substantially harder/easier on larger models.

[1] Ruan, Yangjun, Chris J. Maddison, and Tatsunori B. Hashimoto. "Observational scaling laws and the predictability of langauge model performance." Advances in Neural Information Processing Systems 37 (2024): 15841-15892.

**Questions:**

see weaknesses

---

> ### Author Response · Authors · 2025-11-21
> **Rebuttal**
>
> We thank the reviewer for their thoughtful review of our work. We address the reviewer’s questions below.
>
>
> ### **Q1 Concern about the popularity of pruning**
>
> We sincerely thank the reviewer for their thorough assessment on this point. We split the reviewer’s comments into the following sub-questions.
>
> **Q1-1 Pruning seems less popular compared to quantization. Is it still worth investigating the risk of pruning?**
>
> Yes, we believe it is. Our two major attack target algorithms, Wanda and SparseGPT, both have nearly 1k citations for their papers. Given their overall popularity, we believe there is a significant amount of interest in this field. While it is true that quantization is popular, the security implication of pruning is an interesting field to investigate as an independent topic.
>
> **Q1-2 Pruning is not well-documented in vLLM (i.e., llm-compressor library):**
>
> While, in connection with our argument above, we believe the security risks of pruning are worth investigating on their own; regarding the prevalence of pruning in library documentations and tutorials, we would like to make the following clarifications:
>
> -  [README](https://github.com/vllm-project/llm-compressor/tree/0.8.1) mentions “2:4 Semi-structured and Unstructured Sparsity” and “SparseGPT” as supported formats and algorithms, respectively.
> - While [Getting Started page](https://docs.vllm.ai/projects/llm-compressor/en/stable/getting-started/compress) does not mention pruning (except SparseGPT), this page is linked to the full list of compression schemes, which covers pruning. Also, note that the top page of the document, [About LLM Compressor](https://docs.vllm.ai/projects/llm-compressor/en/stable), also mentions pruning.
> - As the reviewer mentions, the number of tutorials is fewer for pruning than for quantization. We interpret this as reflecting the diversity of quantizations (Weight-only, Weight-and-Activation, KV-Cache, different algorithm types). At a minimum, pruning does exist, as a feature worthy of being covered in tutorials.
>
>
>
> ### **Q2 Can you make Table 1 (utility evaluation) more informative by adding e.g., standard deviation?**
>
> Certainly. We modified Table 1. It now shows the score difference between the base and attacked models with standard deviation over benchmarks, instead of the average over benchmarks. We note that the full results were provided in the Appendix (Tables 8-11). Importantly, the average and standard deviation are both reasonably small, indicating that the score is generally maintained in most of the benchmarks.
>
>
> ### **Q3 Can you ablate $\alpha_{rep} < 1\%$ and also $\alpha_{inj}$?**
>
> Absolutely. Prompted by the reviewer's question, we added the result with $\alpha_{rep} = 0.1\%$ in Figure 2, and the ablation over $\alpha_{inj}$ in Appendix B.1.
>
> Importantly, we find that (i) $\alpha_{rep} = 0.1\%$ is not enough to repair the unpruned model in all three settings, and (ii) our attack is largely robust against the choice of $\alpha_{inj}$.
>
> ### **Q4 How does finetuning or quantization affect the ASR?**
>
> Based on the reviewers' questions, we have now added new experiments in Appendix B1.
>
> For finetuning, we consider the jailbreak setting and finetune the attacked, pruned model by using either the general dataset alone or mixing in security samples (harmful query + refusing response). Here, we find that only the latter setting decreases ASR (achieving near-zero ASR with only 5-step finetuning), whereas finetuning on a general dataset alone does not repair the model. This means that if users want to repair the model through finetuning, they have to identify in which way the model has been attacked, and prepare a corresponding dataset.
>
> For quantization, we tested a rounding-based FP8 quantization and an optimization-based GPTQ (4-bit) quantization. Here, we find that quantization largely preserves ASR (Compared to full precision, quantized ASR is $-7.4 \leq \Delta \leq +14.0$ ).
>
> ### **Q5 Can you elaborate on how the security-aware calibration dataset (Section 6.3) is constructed?**
>
> Certainly. We now clarify this in Section 6.3 with further details in Appendix A.4.
>
> Importantly, we used a subset of the LLM-LAT dataset. We constructed 512 samples to match our main experiment. Throughout the paper, we consider a case where the default maximum sequence length (=2048) is used. Therefore, we concatenate five jailbreaking queries and their refusal responses to construct one sample with roughly the targeted length.
>
> ### **Q6 Can you show results with larger models?**
>
> Absolutely. We conducted an additional experiment using Qwen2.5-14B and 32B models, and provided the result in Appendix B.1. We find that the attack works well for such larger models without specific hyperparameter tunings (ASR of up to $>90$% for each model in each scenario).
>
> We note that although the reviewer asked for the result with 70B models, training of a 70B model is outside the computational capabilities of the authors.

---

> > ### Author Response · Authors · 2025-11-28
> > **Comment by authors**
> >
> > Dear reviewer hEwH,
> >
> > Thank you for taking the time to review our work. We have carefully considered your comments and have done our best to address your concerns. We hope our responses address your concerns, and we would welcome any further questions or clarification you might need.
> >
> > Best regards, Authors

---

### Author Response · Authors · 2025-11-21
**Global Answer to the Reviewers**

We thank all the reviewers for their effort, feedback, and comments.
We are happy to hear that reviewers found that our method is simple/interpretable (hEwH, vaRC), generalizes well (hEwH, QbU6), novel (htX1, vaRC, QbU6), and realistic (htX1, QbU6)


### **Updates to the paper**
---

In light of the reviewers’ feedback, we have now edited the following sections of the paper. The changes are highlighted in blue. We also mention these changes in our individual responses to the corresponding reviewers.

**Figure 1**

We made Figure 1 more detailed so that it more clearly conveys the overall idea of our attack.

**Utility evaluation (Table 1 and 11)**

- In Table 1, instead of showing the average benchmark score, we provide the score change from the base version (in the form of average ± std over five benchmarks).
- We detected some typos (the numbers are wrongly cited in Table 1, and the order of the model name was incorrect in Table 11), so we fixed them accordingly.

**Construction of security-aware calibration (Section 6.3 and A.4)**

We clarified how security-aware calibration is constructed in Section 6.3, with more details and an example in Appendix A.4

**New experimental results**

We added the following results:

- Attacking larger models (Table 5)
- The impact of finetuning and quantization after pruning (Figure 9 and Table 6)
- More ablations on $\alpha_{rep}$ and $\alpha_{inj}$ (Figure 2 and 10)
- Different calibration distribution (Table 7)

---

### Author Response · Authors · 2025-12-02
**Rebuttal Summary**

We would like to thank all reviewers again for their thorough assessment of our paper.
We have received positive assessments from the reviewers **R2(htX1), R3(vaRC), R4(QbU6)** (**8, 6, 8**), while the reviewer **R1(hEwH)** gave the score of **2**.
A major concern of **R1** arises from their fundamental doubt on the popularity of LLM pruning, while noting as a strength that our attack algorithm itself is “correct”.
In contrast, we are happy to see that other reviewers consider our work as practically relevant and realistic.

We summarize our rebuttal as follows:

### **Significance of Our Problem Setting**:

**R1 (Q1)** questioned the popularity of LLM pruning as a whole, and as such, the value of studying its security implications. The reviewer here cites a lack of/scarce documentation of pruning methods in GitHub repositories and tutorials compared to quantization methods. In contrast, we argue that pruning security is an important topic because our target algorithms, Wanda and SparseGPT, are both widely studied (≈1k citations each) and used (integrated in vLLM, 64.3k GitHub stars), and we clarify that existing documentation does in fact include pruning.

Importantly, other reviewers (R2–R4) raised only more specific concerns about the adversarial assumptions. We consider a threat model where a user downloads a compromised, yet seemingly benign model (e.g., from Hugging Face) and locally activates the attack via local pruning with vLLM. Regarding this assumption, we additionally discussed the following with each reviewer:

- **R2 (Q1)**: While users could download already-pruned models (which makes the attack less stealthy), vLLM is explicitly designed to make local pruning easy, so “download then locally prune” is a realistic threat vector.
- **R3 (Q2)**: Users could, in principle, discover the attack after pruning, before deployment. However, exhaustive testing for every type of hidden behavior is expensive. Therefore, post-pruning checks are likely to be incomplete, especially since the model appears secure before pruning.
- **R4 (Q2)**: We clarified that we focus on the widely used unstructured pruning implementations in vLLM to give the attack real-world relevance, and leave structured pruning (with potentially different attack vectors) to future work.

### **Experiments**:

We have added the following experiments in response to the reviewer requests:

**R1 (Q3)**: Additional ablations on $\alpha_{rep}$ and $\alpha_{inj}$ show that too-small $\alpha_{rep}$ values fail (and are unlikely to be chosen by an attacker) and that the attack is largely robust to the choice of $\alpha_{inj}$, supporting our original settings.

**R1 (Q4), R3 (Q1)**: We evaluate quantization and fine-tuning combined with pruning: quantization has little effect on attack success, while fine-tuning can mitigate the attack *only when* using an appropriate security dataset.

**R1 (Q6)**: We demonstrate that the attack is effective on larger (14B and 32B) models.

**R4 (Q3)**: We show the attack still triggers even when the user’s calibration data differs substantially from that used by the attacker.

### **Presentation**:

**R1 (Q2)**: We revised Table 1 to clarify further that pre-pruning utility is not degraded.

**R1 (Q5), R4 (Q1)**: We clarified the “security-aware calibration” ablation and added an example in the Appendix.

**R2 (Q2)**: We improved Figure 1 to convey the overall concept more clearly.

---

### Meta-Review · Area_Chair_rmXc · 2026-01-07

**Summary:**

This paper proposes a new attack on Large Language Models (LLMs) that is triggered only after pruning.

The reviewers appreciate the following strengths of the paper:

S1. The paper reveals a novel threat by proposing a new attack that exploits several standard pruning methods to activate malicious behavior.

S2. The proposed attack is simple, reasonable, and effective across several pruning methods and different pruning levels.

S3. The paper provides extensive experiments.

S4. The paper considers potential countermeasures for the proposed attack.

S5. The paper is well written.

The ratings were mostly positive before the rebuttal. Reviewer QbU6 has explicitly indicated maintaining a positive rating (8). Reviewers htX1 and vaRC provided positive ratings (8, 6) before the rebuttal and are likely to raise or maintain their positive scores. Reviewer hEwH provided a negative rating (2) before the rebuttal, with the primary concern being the popularity of LLM pruning and therefore questioning the practical impact of the proposed attack. The authors have provided additional clarifications on this concern, which appear reasonable.

In summary, the paper has mostly positive ratings, with almost all raised concerns addressed, and can be readily incorporated into the camera-ready version. Thus, this paper is a candidate for final acceptance.

**Reviewer Concerns:**

The reviewers also raised the following major concerns:

W1. Lack of evaluation of additional potential defenses, such as fine-tuning, structured or hardware-specific pruning, and the impact of different distributions of calibration data.

W2. Lack of accuracy in the description of the real-world risk of the threat model.

The authors have addressed most of these concerns during the rebuttal period, as detailed below.

R1. The authors provided new experimental results on the impact of key hyperparameters, quantization and fine-tuning combined with pruning, on larger models (14B and 32B), as well as different calibration datasets.

R2. The authors provided clarifications and a new example for the security-aware calibration ablation.

R3. The authors revised figures and tables as required.

**Reviewer Scores:**

The ratings were mostly positive before the rebuttal. Reviewer QbU6 has explicitly indicated maintaining a positive rating (8). Reviewers htX1 and vaRC provided positive ratings (8, 6) before the rebuttal and are likely to raise or maintain their positive scores. Reviewer hEwH provided a negative rating (2) before the rebuttal, with the primary concern being the popularity of LLM pruning and therefore questioning the practical impact of the proposed attack. The authors have provided additional clarifications on this concern, which appear reasonable.

---

### Decision · Program_Chairs · 2026-01-26

Accept (Poster)